# Exploring the mechanisms linking perceived school climate to negative emotions in adolescents: The mediating roles of social avoidance and distress, and psychological resilience, with physical exercise level as a moderator

Zhen Huang[1], Weisong Chen[2], Bo Peng[2], Haibo Yan[3]*

1 School of physical education, Liaoning Normal University, Dalian, Liaoning, China, 2 School of sports training, Chengdu Sport University, Chengdu, Sichuan, China, 3 School of physical education, Sichuan Normal University, Chengdu, Sichuan, China

* haiboyan0630@163.com

## Abstract

This study investigates the relationships between perceived school climate (PSC) and negative emotions (NEE) in adolescents, incorporating the mediating roles of social avoidance and distress (SAD) and psychological resilience (PSR) and examining the moderating influence of physical exercise level (PEL). Data were collected from 1,507 students in grades 5–9 across China using stratified random sampling. Demographic analysis revealed significant differences across gender, grade, and family location in PSC, SAD, PSR, NEE, and PEL. Structural equation modeling confirmed that PSC directly reduces NEE and exerts additional indirect effects through SAD and PSR, with a sequential chain mediation pathway accounting for the majority of the total effect. Multi-group analysis demonstrated structural invariance across genders. Moderation analysis indicated that PEL amplifies the protective effects of PSC on NEE, with higher levels of physical exercise strengthening this association. These findings provide comprehensive insights into the interplay of environmental, psychological, and behavioral factors, offering valuable implications for targeted educational and policy interventions.

## 1. Introduction

Adolescence is a formative period characterized by significant emotional and psychological development, during which external environments, such as school climate, play a critical role in shaping mental health outcomes [1]. Perceived school climate—adolescents' subjective evaluations of the school environment, including aspects of safety, support, and inclusiveness—has emerged as a key factor influencing

**Data availability statement:** All relevant data are within the manuscript and its Supporting Information files.

**Funding:** This study was supported by the Major Project of the Sichuan Key Research Base of Philosophy and Social Sciences (Grant No. SC21EZD006), titled "Research on the Collaborative Development of School Physical Education in the Chengdu-Chongqing Economic Circle from the Perspective of Integration of Sports and Education." The funders had no role in the study design, data collection and analysis, decision to publish, or preparation of the manuscript.

**Competing interests:** The authors have declared that no competing interests exist.

emotional well-being [2,3]. A positive perceived school climate has been consistently associated with lower levels of negative emotions, such as anxiety and depression, while a negative school climate often exacerbates these issues [4–6]. Despite its significance, the mechanisms and contextual factors shaping the relationship between perceived school climate and emotional outcomes remain insufficiently explored.

One potential explanation lies in the mediating roles of social avoidance and distress, and psychological resilience. Social avoidance and distress, defined as discomfort in and avoidance of social interactions, have been identified as significant risk factors for emotional disturbances, such as anxiety and depression [7,8]. Adolescents experiencing a negative school climate are more likely to withdraw from social interactions, increasing isolation and emotional distress [9,10]. Conversely, a supportive school climate fosters trust and positive peer interactions, thereby reducing social avoidance and alleviating distress [11,12]. Psychological resilience, defined as the ability to adapt positively in the face of adversity, represents another critical mechanism linking school climate to emotional well-being [13–15]. A positive school climate strengthens resilience by offering supportive relationships and opportunities for personal growth, which in turn mitigates the impact of stressors on negative emotions [16]. Importantly, these mediators may function sequentially, with reduced social avoidance creating opportunities for resilience to develop, which ultimately enhances emotional stability [17].

While these internal psychological mechanisms highlight pathways linking perceived school climate to emotional health, external behavioral factors, such as physical exercise level, may also play a significant role. Physical exercise, widely recognized for its psychological benefits, including stress reduction and improved emotional regulation, may moderate the direct relationship between perceived school climate and negative emotions [18,19]. For instance, adolescents who engage in regular physical activity may experience enhanced resilience and emotional regulation, making them less vulnerable to the emotional challenges of a negative school climate [19]. Similarly, in the context of a positive school climate, physical exercise may amplify its protective effects on emotional health, creating an additive benefit. Examining physical exercise as a moderating factor provides essential insights into how environmental and behavioral factors interact to influence adolescent emotional outcomes.

This study aims to examine the complex relationships among perceived school climate, social avoidance and distress, psychological resilience, and negative emotions in adolescents. Specifically, it seeks to explore the mediating roles of social avoidance and distress, and psychological resilience, as well as the moderating effect of physical exercise level on the direct relationship between perceived school climate and negative emotions. By integrating these mediating and moderating mechanisms into a unified framework, this research advances theoretical insights into the environmental, psychological, and behavioral factors shaping emotional outcomes in adolescents and provides practical guidance for designing school-based interventions to enhance psychological well-being.

## 2. Literature review and research hypotheses

### 2.1. Perceived school climate and negative emotions

Perceived school climate, encompassing teacher support, peer support, and opportunities for autonomy, is a widely recognized determinant of adolescent emotional well-being [20,21]. Positive perceptions of the school climate are consistently linked to lower levels of negative emotions, including depression, anxiety, and stress [4]. Teacher support creates a secure foundation for emotional growth by providing consistent encouragement and constructive feedback [22,23]. Peer support, as a critical aspect of school climate, fosters social integration and reduces feelings of alienation, both of which are protective factors against emotional disturbances [21]. Additionally, opportunities for autonomy promote self-efficacy and agency, which further contribute to psychological well-being [24–26].

However, while the direct association between perceived school climate and negative emotions has been extensively studied, the lack of integration between direct and indirect pathways limits our understanding of how school climate impacts emotional outcomes. Specifically, previous research often treats perceived school climate as a monolithic construct with direct effects, neglecting its interaction with behavioral factors or its influence through psychological mechanisms. This oversimplification fails to capture the nuanced ways in which adolescents experience and respond to their school environment.

This study addresses these gaps by situating the direct relationship between perceived school climate and negative emotions within a broader framework that includes both mediating and moderating mechanisms. By integrating the roles of social avoidance and distress, psychological resilience, and physical exercise level, this research moves beyond traditional linear models and offers a more comprehensive perspective. This approach not only elucidates the direct relationship but also examines how behavioral and psychological factors interplay to shape emotional outcomes in adolescents.

### 2.2. Mediating variables in the relationship between perceived school climate and negative emotions

**2.2.1. Social avoidance and distress.** Social avoidance and distress, involving discomfort in and withdrawal from social interactions, are significant risk factors for negative emotional outcomes, particularly during adolescence, a period marked by heightened sensitivity to social experiences [27–29]. A negative perceived school climate often amplifies these tendencies by fostering an environment of exclusion, mistrust, or even hostility, leading adolescents to withdraw socially and experience elevated levels of emotional distress [12,30]. This withdrawal not only intensifies feelings of loneliness and anxiety but also creates a self-perpetuating cycle that reinforces negative emotions.

Conversely, a positive perceived school climate, characterized by strong teacher and peer support, can interrupt this cycle by fostering trust and inclusivity [31]. Such an environment encourages adolescents to engage in meaningful social interactions, which can mitigate avoidance behaviors and alleviate emotional distress. While existing research recognizes the role of social avoidance in adolescent emotional health, it often treats it as an outcome rather than a mechanism linking perceived school climate to negative emotions. This study advances the literature by positioning social avoidance and distress as mediators, shedding light on the social pathways through which school environments influence emotional outcomes.

**2.2.2. Psychological resilience.** Psychological resilience, defined as the capacity to adapt positively and recover from adversity, plays a central role in buffering adolescents against emotional challenges [32–34]. A positive perceived school climate promotes resilience by providing emotional support, fostering a sense of belonging, and creating opportunities for personal and social development [35,36]. These factors enable adolescents to regulate their emotions more effectively, recover from setbacks, and maintain psychological stability, even in the face of stressors.

In contrast, a negative perceived school climate may hinder the development of resilience by creating an environment of fear or exclusion, depriving adolescents of critical emotional and relational resources [37]. Without sufficient resilience, adolescents are less equipped to manage stress, leading to the accumulation of negative emotions. While previous

studies have established the protective role of resilience, this study uniquely situates resilience as a mediator, exploring how supportive school environments can indirectly reduce negative emotions by enhancing adaptive capacity.

### 2.3. Chain mediation effect

Perceived school climate, as a key environmental factor, not only directly influences adolescents' emotional outcomes but also shapes their social behaviors and psychological resources in a sequential manner [21]. A supportive and inclusive school climate fosters trust, a sense of belonging, and opportunities for meaningful social interactions, which reduce the likelihood of social avoidance and distress [31,38]. By providing a safe and encouraging environment, positive school climates motivate adolescents to engage with peers and build supportive relationships [39]. Conversely, a negative perceived school climate characterized by exclusion or lack of support tends to reinforce feelings of isolation, leading to withdrawal behaviors and heightened emotional distress [40].

The reduction in social avoidance and distress facilitated by a positive school climate creates opportunities for adolescents to strengthen their psychological resources, particularly psychological resilience [41,42]. Resilience enables individuals to adapt to adversity and recover from emotional setbacks, acting as a crucial buffer against negative emotions [43,44]. For instance, frequent and meaningful social interactions provide adolescents with access to emotional support and coping mechanisms, promoting the accumulation of adaptive capacities over time. On the other hand, prolonged social avoidance and distress limit access to these resources, impairing adolescents' ability to manage stress effectively and exacerbating negative emotional states [45].

Psychological resilience, as the culmination of adaptive capacities developed through positive social experiences, serves as a critical safeguard for emotional health [33]. Adolescents with high resilience exhibit greater emotional stability when faced with challenges, while those with low resilience are more likely to experience the accumulation of negative emotions over time [44,46,47]. A positive perceived school climate indirectly reduces negative emotions by fostering resilience, thereby providing adolescents with a vital protective mechanism against stress and adversity [35].

This study uniquely integrates social avoidance and distress, and psychological resilience into a chain mediation model to explore their sequential roles in the relationship between perceived school climate and negative emotions. By connecting social behaviors (social avoidance and distress) with psychological adaptability (psychological resilience), this model offers a nuanced perspective that captures the dynamic pathways through which school climate impacts emotional health. Unlike existing studies that focus on isolated mediating factors, this approach highlights the interconnected nature of social and psychological processes, contributing to a deeper theoretical understanding of adolescent emotional development.

### 2.4. Moderating effect

Physical exercise, as a widely recognized positive lifestyle choice, contributes not only to physical health but also to psychological well-being through its impact on emotional regulation mechanisms [48,49]. Extensive research has demonstrated that physical exercise improves neurotransmitter functioning, such as increasing endorphin and serotonin levels, which are closely associated with reduced anxiety and depression [50–52]. For adolescents, regular physical activity strengthens their ability to cope with stress and lowers the likelihood of experiencing negative emotions, highlighting its critical role in promoting emotional stability [53].

The relationship between perceived school climate and negative emotions is not uniform and may vary depending on behavioral factors such as physical exercise. In a positive perceived school climate, students who engage in regular physical activity may benefit from enhanced emotional regulation, which further reduces negative emotions by improving their perception of the school environment [54,55]. This indicates that physical exercise amplifies the emotional benefits of a supportive school climate, creating a synergistic effect on well-being. On the other hand, for students in a negative perceived school climate, physical exercise may serve as a buffer, mitigating the adverse emotional effects associated

with a less supportive school environment [56]. This buffering effect highlights physical exercise as a key protective factor for emotionally vulnerable students.

Moreover, the intensity and frequency of physical exercise significantly influence its moderating effect. High-frequency or moderate-to-high-intensity exercise has been shown to yield more substantial psychological benefits in alleviating negative emotions compared to low-frequency or low-intensity exercise [57,58]. These findings underscore the importance of encouraging consistent and sufficiently intense physical activity to optimize its emotional regulation effects [59].

By examining the moderating role of physical exercise, this study provides an integrated framework that connects environmental, psychological, and behavioral factors. Unlike previous studies that have largely treated physical exercise as an independent predictor of emotional outcomes, this research highlights its interactive role in shaping the relationship between perceived school climate and negative emotions. This perspective contributes to a more nuanced understanding of emotional regulation mechanisms and offers practical implications for designing holistic intervention strategies that combine school environment improvements with physical activity promotion.

### 2.5. Theoretical foundations

**2.5.1. Stress-buffering hypothesis: Immediate social responses to school climate.** The Stress-Buffering Hypothesis contends that supportive social environments alleviate the deleterious effects of stress, whereas unsupportive settings exacerbate stress-related outcomes [60,61]. In the context of school climate, a negative environment—marked by inadequate teacher support, peer rejection, or limited autonomy—constitutes an immediate stressor [62]. Adolescents exposed to such environments often experience heightened levels of anxiety and discomfort, leading to social avoidance and distress (SAD). This response can be understood as an initial coping strategy that, while reducing immediate exposure to further stressors, inadvertently limits opportunities for positive social engagement and emotional validation. Over time, this maladaptive response not only intensifies feelings of isolation but also contributes directly to negative emotional outcomes. Thus, the Stress-Buffering Hypothesis provides a conceptual foundation for positioning SAD as an early mediator in the pathway from PSC to NEE.

**2.5.2. Developmental cascade theory: Shaping adaptive capacity over time.** Complementing the immediate effects outlined by the Stress-Buffering Hypothesis, the Developmental Cascade Theory emphasizes how early experiences initiate a sequence of developmental processes that influence later outcomes [63,64]. According to this theory, early social experiences—whether positive or negative—cumulatively shape an individual's psychological resources [65,66]. In our model, prolonged exposure to a negative school climate results in sustained social avoidance and distress, which in turn impedes the development of psychological resilience (PSR). Resilience is not an innate, fixed trait; rather, it is an adaptive capacity that evolves over time as adolescents learn to cope with adversity through their social interactions. Positive school climates, by contrast, provide opportunities for constructive engagement that foster resilience, enabling adolescents to better regulate their emotions in the face of stress [41]. Therefore, Developmental Cascade Theory justifies the sequential ordering of our mediators, where SAD, as an initial maladaptive reaction, disrupts the development of resilience, thereby leading to greater negative emotional outcomes.

**2.5.3. Integrating the moderating role of physical exercise.** In addition to these mediating processes, our model introduces physical exercise (PEL) as a moderator, based on the Stress-Buffering Hypothesis and Developmental Cascade Theory. Unlike resilience, which develops as a response to prior experiences, physical exercise is an external behavioral factor that can actively influence how adolescents respond to environmental stressors [67]. The Stress-Buffering Hypothesis suggests that engaging in regular physical activity mitigates the psychological impact of negative experiences by enhancing physiological stress resilience and emotional regulation [68,69]. Meanwhile, Developmental Cascade Theory highlights the long-term benefits of physical activity in shaping adaptive coping mechanisms and reducing emotional vulnerability [70].

                                                                    

Empirical research supports the notion that physical exercise modulates the effects of environmental influences on emotional well-being. Regular engagement in physical activity promotes mood-regulating neurotransmitter release (e.g., endorphins, serotonin), strengthens self-efficacy, and fosters better stress adaptation, which can either amplify the benefits of a positive school climate or buffer the negative effects of an unfavorable one [71,72]. Unlike a mediating role, which would imply that school climate directly influences engagement in physical activity, we propose that exercise operates as a protective factor that interacts with school climate to shape emotional outcomes. This conceptualization aligns with findings that adolescents with higher levels of physical activity tend to exhibit greater emotional resilience, regardless of their school climate perceptions, supporting the rationale for its moderating role in our model.

**2.5.4. Synthesis and theoretical implications.** By integrating the Stress-Buffering Hypothesis and Developmental Cascade Theory, our theoretical foundation elucidates the following sequential process:

(1) Immediate impact of school climate: A negative school climate triggers social avoidance and distress as an immediate response, consistent with stress-buffering mechanisms.

(2) Long-term adaptation: The persistence of social avoidance and distress limits opportunities for the development of psychological resilience, as posited by developmental cascade processes.

(3) Emotional outcomes: The resulting deficit in resilience diminishes adolescents' capacity to effectively manage stress, culminating in increased negative emotions.

(4) Behavioral moderation: Physical exercise moderates the direct relationship between school climate and negative emotions, highlighting its role in enhancing or buffering emotional responses.

This integrative theoretical framework provides a solid foundation for our model's structure, offering support for the proposed directional relationships, the sequential mediation process, and the moderating influence of physical exercise.

## 2.6. Theoretical model and hypotheses

Based on these theoretical foundations, the study posits the following hypotheses:

**Hypothesis 1 (H1):** Perceived school climate is negatively associated with adolescents' negative emotions.

**Hypothesis 2 (H2):** Social avoidance and distress mediate the relationship between perceived school climate and negative emotions.

**Hypothesis 3 (H3):** Psychological resilience mediates the relationship between perceived school climate and negative emotions.

**Hypothesis 4 (H4):** Social avoidance and distress and psychological resilience jointly form a chain mediation effect between perceived school climate and negative emotions.

**Hypothesis 5 (H5):** Physical exercise level moderates the relationship between perceived school climate and negative emotions.

[Fig 1] illustrates the proposed theoretical model, which integrates the direct effect of perceived school climate on negative emotions, the sequential mediation via social avoidance/distress and psychological resilience, and the moderating role of physical exercise.

## 2.7. Contributions of this study

This study contributes to the existing body of research by addressing critical gaps and advancing theoretical understanding:

1. Integration of multiple mechanisms: By incorporating mediating and moderating processes, the model offers a holistic approach to understanding how school climate impacts adolescent emotional health.

2. Exploration of chain mediation: The sequential influence of social avoidance, distress, and resilience highlights the dynamic interplay between social and psychological processes, which has been overlooked in previous studies.

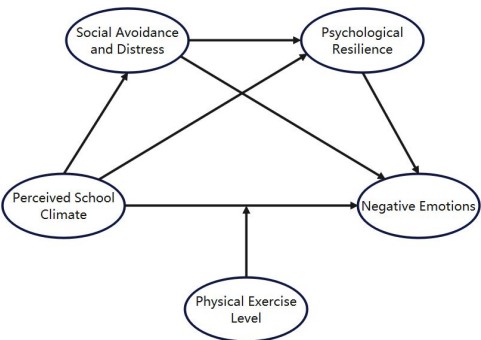

**Fig 1. Theoretical model of the relationship between perceived school climate and negative emotions: The mediating roles of social avoidance and distress and psychological resilience and the moderating role of physical exercise level.**

3. Inclusion of behavioral moderation: The role of physical exercise as a moderator enriches the framework by emphasizing the interactive nature of environmental and behavioral factors.

## 3. Materials and methods

### 3.1. Participants and data

**3.1.1. Sample size determination.** To ensure the statistical robustness and reliability of this study, a sample size estimation was conducted using G*Power 3.1 software, a widely recognized tool in social science research. The power analysis focused on a multiple regression model involving five predictors: Perceived school climate (independent variable), negative emotions (dependent variable), social avoidance and distress (mediator), psychological resilience (mediator), and physical exercise Level (moderator). Assuming a medium effect size ($f^2 = 0.15$), a significance level of $\alpha = 0.05$, and a desired statistical power of $1 - \beta = 0.80$, the analysis indicated that a minimum sample size of 327 participants would be required to detect statistically significant effects.

In addition to the G*Power estimation, the participant-to-item ratio guideline for survey-based studies was considered. This guideline suggests a ratio of 5–10 participants per survey item. The survey instrument used in this study includes a total of 107 items: Perceived school climate scale (25 items), negative emotions scale (21 items), social avoidance and distress scale (28 items), psychological resilience scale (27 items), and physical exercise level scale (3 items). Based on this guideline, a recommended sample size range of 535–1,070 participants was established.

To ensure sufficient statistical power, generalizability, and robustness across demographic subgroups, 1,600 questionnaires were distributed to adolescents in middle and high schools. This sample size exceeds both the G*Power requirement and the participant-to-item ratio recommendation, ensuring that the findings of this study are statistically robust and generalizable across diverse adolescent populations.

**3.1.2. Participant selection process.** This study focused on students in grades 5–9, including upper elementary school (grades 5 and 6) and middle school (grades 7–9). Lower elementary students (grades 1–4) were excluded due to their limited ability to comprehend the survey. Stratified random sampling was employed to ensure representation across grades, gender, and family location (urban and rural).

Schools were randomly selected from multiple regions in China. Within each selected school, one or two classes per grade were randomly chosen, and all students in these classes were invited to participate. Participants had to meet the following inclusion criteria: being enrolled in grades 5–9, capable of independently completing the survey, and regularly engaging in physical exercise level. Invalid responses, such as incomplete answers or fixed response patterns, were excluded during data cleaning.

**3.1.3 Data collection methods.** To ensure validity, reliability, and consistency, data collection followed standardized procedures:

**Training of data collectors**: All research assistants underwent comprehensive training, including the study's objectives, the importance of ethical practices, and standardized procedures for survey administration. This ensured consistent instructions provided to participants and minimized potential biases.

**Questionnaire administration**: Surveys were administered during regular school hours in classroom settings to minimize disruption to participants' routines. Research assistants distributed the questionnaires and provided clear instructions. Students were encouraged to complete the survey independently and honestly, without discussing responses with peers. Completed questionnaires were collected immediately to ensure data integrity.

**Control measures to reduce bias:** Given that this study relied on self-reported data, several measures were implemented to minimize common method bias, social desirability effects, and response inaccuracies: (1) Anonymity Assurance: Participants were explicitly informed that their responses would remain anonymous and confidential, reducing pressure to provide socially desirable answers. (2) Validated Psychological Scales: All self-report measures were selected based on previous validation in adolescent populations, ensuring high reliability and construct validity. (3) Balanced Question Framing: Items were neutral and non-leading, minimizing biased responses. (4) Standardized Instructions: Participants received standardized verbal and written instructions, emphasizing the importance of honest and accurate responses. (5) Response Consistency Checks: We conducted post-survey data screening for inconsistent or extreme responses to improve data quality. (6) Missing Data Handling: All samples with missing data were completely removed from the dataset, ensuring that only fully completed responses were analyzed. This approach helps maintain data integrity and prevents potential biases from imputation methods.

**Confidentiality and informed consent**: Ethical approval for this study was obtained from the Ethics Committee of Chengdu Sport University (Institutional Review Board, IRB Approval No. CTYLL2024014), in compliance with the Declaration of Helsinki and national ethical standards. The Ethics Committee, serving as the Institutional Review Board (IRB), reviewed and approved the study protocol, including the procedure for obtaining verbal informed consent as appropriate for this research. Prior to participation, both the students and their guardians were provided with a clear and detailed verbal explanation of the study's objectives, procedures, potential risks, and their rights, including the right to withdraw at any time without consequence. To ensure transparency and accountability, verbal consent was witnessed by two independent school administrators and documented in written logs. These logs recorded the participant's ID code (e.g., SS-2024–001), the date of consent, and signatures from both witnesses confirming the participant's voluntary agreement. Confidentiality and anonymity were rigorously maintained throughout the study. No personally identifiable information was collected, and all data were fully anonymized and securely stored. All consent documentation, including written logs and witness verification records, was encrypted using SHA-256 standards and archived in the university's social science data repository for a mandatory 15-year preservation period, in accordance with national academic ethics regulations.

**3.1.4 Data processing.** The survey was conducted from April 1 to May 31, 2024. A total of 1,600 questionnaires were distributed to students in grades 5–9. After excluding invalid responses due to missing data, fixed response patterns, or inconsistencies, 1,507 valid questionnaires were retained, resulting in an effective response rate of 94.19%.

Invalid questionnaires were identified and excluded based on predefined criteria, such as incomplete answers and uniform response patterns. Data cleaning ensured the quality and completeness of the dataset, providing a robust foundation for subsequent analysis. Demographic information of the respondents is presented in **Table 1**.

## 3.2. Measurement tools

Standardized tools were employed to assess the independent variable, dependent variable, mediating variables, and moderating variable:

**Perceived school climate**: Assessed using the scale developed by Jia et al. (2009), measuring students' perceptions of their school environment [73]. The scale includes 25 items scored on a 4-point Likert scale across three

**Table 1. The sample information.**

| Basic information | Category | Frequency | Percentage | Cumulative percentage |
|---|---|---|---|---|
| Gender | Male | 952 | 63.17 | 63.17 |
| | Female | 555 | 36.83 | 100 |
| Grade | Grade 5 | 318 | 21.1 | 21.1 |
| | Grade 6 | 297 | 19.71 | 40.81 |
| | Grade 7 | 305 | 20.24 | 61.05 |
| | Grade 8 | 296 | 19.64 | 80.69 |
| | Grade 9 | 291 | 19.31 | 100 |
| Family location | Urban | 742 | 49.24 | 49.24 |
| | Rural | 765 | 50.76 | 100 |

dimensions: teacher support (7 items), peer support (13 items), and autonomy opportunities (5 items). Higher scores indicate a more positive perception of the school climate.

**Negative emotions**: Measured using the Chinese version of the Depression-Anxiety-Stress Scale (DASS-21), validated by Gong et al. (2010) [74]. The scale includes 21 items scored on a 4-point Likert scale across three dimensions: depression (7 items), anxiety (7 items), and stress (7 items). Higher scores indicate higher levels of negative emotions.

**Social avoidance and distress**: Measured using the Social Avoidance and Distress Scale developed by Watson and Friend (1969) [75]. The scale includes 28 items scored on a 2-point scale (1 = No, 2 = Yes) across two dimensions: social avoidance (14 items) and social distress (14 items). Higher scores indicate greater social discomfort and avoidance.

**Psychological resilience** was assessed using the Resilience Scale for Adolescents, developed and validated by Hu and Gan (2008) [76]. This positive variable consists of 27 items scored on a 5-point Likert scale across five dimensions: goal focus (5 items), emotional control (6 items), positive cognition (4 items), family support (6 items), and interpersonal assistance (6 items). Higher scores represent greater psychological resilience, indicating better adaptive capacity in the face of challenges.

**Physical exercise level** was assessed using a scale developed by Liang Deqing et al. [77], which includes 3 items: exercise intensity, duration, and frequency. The intensity and duration of physical exercise were rated on a 5-point Likert scale, ranging from 1 (very poor) to 5 (excellent), while the frequency was rated from 0 (never) to 4 (always). The scores for intensity, duration, and frequency were multiplied to calculate a total score, ranging from 0 to 100. Previous studies have demonstrated that this scale has high reliability and validity.

Reverse-scored items across the scales were adjusted to ensure consistency in positive scoring. The use of validated tools ensures the reliability and validity of the measurements in this study.

### 3.3. Data analysis procedure

Data were analyzed using SPSS 26.0 and AMOS 24.0. Descriptive statistics (mean, standard deviation) were calculated to understand the distribution of key variables. Pearson's correlation analysis was conducted to examine the relationships among Perceived school climate, negative emotions, social avoidance and distress, psychological resilience, and physical exercise level. To identify potential subgroup differences, independent sample t-tests and one-way ANOVA were used to test for variations across demographic variables such as gender, grade, and family location.

Harman's single-factor test was used to assess common method bias, ensuring that the observed relationships were not influenced by measurement artifacts. Structural equation modeling (SEM) was performed in AMOS 24.0 to examine the chain mediating roles of social avoidance and distress and psychological resilience in the relationship between perceived school climate and negative emotions. Model fit indices (e.g., $\chi^2/df$, CFI, RMSEA) were used to validate the model, and bootstrapping with 2,000 resamples was employed to generate confidence intervals for indirect effects.

 

Additionally, multi-group invariance testing was conducted to evaluate whether the structural relationships between perceived school climate and negative emotions were consistent across gender groups. Changes in comparative fit indices (ΔCFI < 0.01) were used to determine invariance between gender groups.

Finally, moderation analysis was conducted using the PROCESS macro for SPSS to test the moderating effect of physical exercise level on the relationship between perceived school climate (independent variable) and negative emotions (dependent variable). Interaction effect models in the PROCESS macro were used to evaluate how physical exercise level influences the strength of the relationship between perceived school climate and negative emotions.

## 4. Result

### 4.1. Common method bias test

To assess the potential impact of common method bias, a principal component analysis was conducted using Harman's single-factor test. Fourteen factors with eigenvalues greater than 1 were extracted, with the largest factor accounting for 25.65% of the total variance, which is below the critical threshold of 40%. This indicates that common method bias is not a significant concern in this study, suggesting minimal interference with subsequent analyses.

### 4.2. Descriptive statistics, reliability, and construct validity of the measurement model

The descriptive statistics and internal consistency reliability for the key variables are summarized in **Table 2**. The mean (M) and standard deviation (SD) indicate moderate to high variability across constructs. Perceived school climate had a mean of 3.077 (SD = 0.534), while social avoidance and distress reported a mean of 1.306 (SD = 0.299). Psychological resilience had a mean of 3.563 (SD = 0.666), and negative emotions exhibited a mean of 2.574 (SD = 0.546). Physical exercise level showed the highest variability, with a mean of 37.071 (SD = 30.210).

The internal consistency reliability, assessed using Cronbach's alpha (α), demonstrated strong reliability for all constructs. Values ranged from 0.923 (negative emotions) to 0.950 (social avoidance and distress), exceeding the commonly accepted threshold of 0.70, confirming the reliability of the scales.

The confirmatory factor analysis (CFA) results in **Table 2** indicate robust construct validity. Fit indices for all variables met the recommended thresholds for a good model fit. Comparative fit index (CFI) values ranged from 0.951 to 0.976, and Tucker-Lewis index (TLI) values ranged from 0.946 to 0.974, both surpassing the acceptable cutoff of 0.90. The standardized root mean square residual (SRMR) values were below 0.08, with the lowest value being 0.024 for psychological resilience. Additionally, the root mean square error of approximation (RMSEA) values, including the 90% confidence intervals, ranged from 0.034 to 0.050, indicating excellent model fit across constructs.

To further verify the construct validity, alternative factor structures were compared, as shown in **Table 3**. The four-factor model (perceived school climate, social avoidance and distress, psychological resilience, and negative emotions) demonstrated the best fit, with $\chi^2$ = 164.396, *df* = 59, CFI = 0.986, TLI = 0.982, SRMR = 0.020, and RMSEA = 0.034 (90% CI = 0.028–0.041). Comparatively, the three-factor, two-factor, and one-factor models showed significantly poorer fit, with increasing $\chi^2$ values and decreasing CFI and TLI values, alongside higher SRMR and RMSEA values. Notably, the

**Table 2. Descriptive statistics, internal consistency reliability, and fit indices for confirmatory factor analysis (CFA) of key variables.**

| Variable | M | SD | α | CFI | TLI | SRMR | RMSEA (90%CI) |
|---|---|---|---|---|---|---|---|
| Perceived school climate | 3.077 | 0.534 | 0.943 | 0.951 | 0.946 | 0.027 | 0.050 (0.048–0.053) |
| Social avoidance and distress | 1.306 | 0.299 | 0.950 | 0.976 | 0.974 | 0.025 | 0.034 (0.031–0.037) |
| Psychological resilience | 3.563 | 0.666 | 0.941 | 0.970 | 0.967 | 0.024 | 0.037 (0.034–0.039) |
| Negative emotions | 2.574 | 0.546 | 0.923 | 0.972 | 0.969 | 0.025 | 0.039 (0.035–0.042) |
| Physical exercise level | 37.071 | 30.210 | 0.839 | – | – | – | – |

**Table 3. Comparative fit indices for alternative factor structures of key constructs.**

| Model | Factor | χ 2 | df | △χ 2 (△df) | CFI | TLI | SRMR | RMSEA (90%CI) |
|---|---|---|---|---|---|---|---|---|
| Four-factor model | PSC, SAD, PSR, NEE | 164.396 | 59 | – | 0.986 | 0.982 | 0.020 | 0.034 (0.028–0.041) |
| Three-factor model | PSC+ SAD, PSR, NEE | 760.357 | 62 | 595.961 (3) | 0.908 | 0.885 | 0.059 | 0.086 (0.081–0.092) |
| Two-factor model | PSC+ SAD+ PSR, NEE | 1092.029 | 64 | 927.633 (5) | 0.865 | 0.836 | 0.069 | 0.103 (0.098–0.109) |
| One-factor model | PSC+ SAD+ PSR+ NEE | 1186.092 | 65 | 1021.696 (6) | 0.853 | 0.824 | 0.070 | 0.107 (0.102–0.112) |

PSC, perceived school climate. NEE, negative emotions. SAD, social avoidance and distress. PSR, psychological resilience. All △χ 2 passed the significance test at 0.05 level.

three-factor model (χ² = 760.357, df = 62) exhibited acceptable fit indices but was significantly inferior to the four-factor model, as indicated by the Δχ² tests. The two-factor and one-factor models performed even worse, failing to meet acceptable thresholds for several fit indices.

Overall, the results support the proposed four-factor model as the most appropriate representation of the constructs. The high internal consistency reliability and excellent fit indices confirm the adequacy of the measurement model, providing a robust foundation for subsequent analyses.

### 4.3 Correlation analysis among key variables

The correlation analysis among key variables is presented in **Fig 2**. Perceived school climate (PSC) showed a significant negative correlation with negative emotions (NEE) (r = −0.42, $p$ < 0.001) and social avoidance and distress (SAD) (r = −0.40, $p$ < 0.001), while demonstrating a positive correlation with psychological resilience (PSR) (r = 0.42, $p$ < 0.001). A weaker but significant positive association was observed between PSC and physical exercise level (PEL) (r = 0.28, $p$

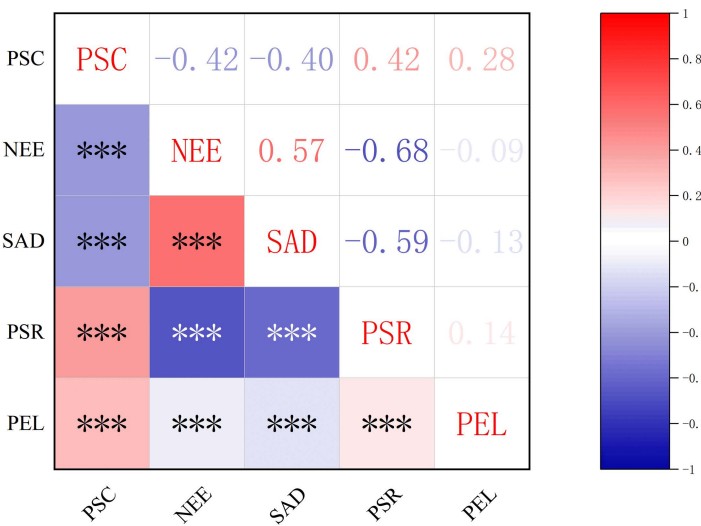

\* p<=0. 05　\*\* p<=0. 01　\*\*\* p<=0. 001

**Fig 2. Correlation matrix of key variables.** PSC, perceived school climate. NEE, negative emotions. SAD, social avoidance and distress. PSR, psychological resilience. PEL, physical exercise level.

< 0.001). SAD was strongly positively correlated with NEE (r = 0.57, *p* < 0.001) and negatively correlated with PSR (r = -0.59, *p* < 0.001), but its correlation with PEL was non-significant (r = -0.13, *p* > 0.05). PSR was negatively associated with NEE (r = −0.68, *p* < 0.001) and weakly positively correlated with PEL (r = 0.14, *p* < 0.05). The correlation between PEL and NEE was not statistically significant (r = -0.09, *p* > 0.05). These results confirm the interrelationships among the constructs and support the hypothesized relationships in the proposed model.

## 4.4 Differences in key variables among demographic groups of adolescents

The descriptive statistics and differences in key variables across demographic groups are presented in **Table 4**. Analyses were conducted based on gender, grade, and family location.

For gender, significant differences were observed across all key variables. Female students reported a higher perceived school climate (M = 3.180, SD = 0.507) compared to males (M = 3.016, SD = 0.540, t = −5.897, *p* < 0.001). Female students also reported higher levels of social avoidance and distress (M = 1.329, SD = 0.299) and negative emotions (M = 2.629, SD = 0.536) than males, with t values of −2.340 (*p* = 0.019) and −2.961 (*p* = 0.003), respectively. Conversely, male students demonstrated higher levels of psychological resilience (M = 3.608, SD = 0.667) and physical exercise (M = 39.246, SD = 29.965) compared to females, with t values of 3.452 (*p* = 0.001) and 3.675 (*p* < 0.001), respectively.

Regarding grade, significant differences were found across all variables. Grade 5 students reported the highest perceived school climate (M = 3.199, SD = 0.521), social avoidance and distress (M = 1.232, SD = 0.267), and psychological resilience (M = 3.668, SD = 0.677). In contrast, Grade 9 students reported the lowest perceived school climate (M = 2.897, SD = 0.514), resilience (M = 3.460, SD = 0.647), and the highest negative emotions (M = 2.662, SD = 0.519). Physical exercise levels also varied significantly, with Grade 5 students engaging in the highest levels of exercise (M = 41.399, SD = 30.165), while Grade 9 students reported the lowest levels (M = 29.000, SD = 26.888). The overall grade differences were significant, with F values ranging from 3.857 to 13.205 (all *p* < 0.05).

For family location, students from rural areas exhibited higher perceived school climate (M = 3.136, SD = 0.516) and physical exercise levels (M = 41.013, SD = 29.993) compared to urban students (M = 3.015, SD = 0.545; M = 33.007, SD = 30.005), with t values of −4.415 (*p* < 0.001) and −5.188 (*p* < 0.001), respectively. Rural students also reported lower

**Table 4. Descriptive statistics and differences across demographic variables for key variables.**

| Demographic variables | Category | Perceived school climate | | Social avoidance and distress | | Psychological resilience | | Negative emotions | | Physical exercise level | |
|---|---|---|---|---|---|---|---|---|---|---|---|
| | | M | SD | M | SD | M | SD | M | SD | M | SD |
| Gender | Male | 3.016 | 0.540 | 1.292 | 0.298 | 3.608 | 0.667 | 2.542 | 0.550 | 39.246 | 29.965 |
| | Female | 3.180 | 0.507 | 1.329 | 0.299 | 3.485 | 0.657 | 2.629 | 0.536 | 33.341 | 30.290 |
| | *t* | −5.897 | | −2.340 | | 3.452 | | −2.961 | | 3.675 | |
| | *P* | 0.000 | | 0.019 | | 0.000 | | 0.003 | | 0.000 | |
| Grade | Grade 5 | 3.199 | 0.521 | 1.232 | 0.267 | 3.668 | 0.677 | 2.484 | 0.556 | 41.399 | 30.165 |
| | Grade 6 | 3.068 | 0.540 | 1.299 | 0.293 | 3.539 | 0.673 | 2.584 | 0.567 | 37.741 | 30.743 |
| | Grade 7 | 3.101 | 0.527 | 1.312 | 0.306 | 3.575 | 0.666 | 2.559 | 0.550 | 38.328 | 30.045 |
| | Grade 8 | 3.104 | 0.525 | 1.319 | 0.301 | 3.562 | 0.652 | 2.591 | 0.525 | 38.389 | 31.681 |
| | Grade 9 | 2.897 | 0.514 | 1.373 | 0.312 | 3.460 | 0.647 | 2.662 | 0.519 | 29.000 | 26.888 |
| | *F* | 13.205 | | 8.864 | | 3.857 | | 4.246 | | 7.252 | |
| | *P* | 0.000 | | 0.000 | | 0.004 | | 0.002 | | 0.001 | |
| Family location | Urban | 3.015 | 0.545 | 1.321 | 0.304 | 3.618 | 0.648 | 2.583 | 0.546 | 33.007 | 30.005 |
| | Rural | 3.136 | 0.516 | 1.290 | 0.293 | 3.509 | 0.679 | 2.566 | 0.547 | 41.013 | 29.903 |
| | *t* | −4.415 | | 2.023 | | 3.212 | | 0.607 | | −5.188 | |
| | *P* | 0.000 | | 0.043 | | 0.001 | | 0.544 | | 0.000 | |

social avoidance and distress (M = 1.290, SD = 0.293) compared to urban students (M = 1.321, SD = 0.304, t = 2.023, *p* = 0.043). No significant differences were observed in negative emotions (*p* = 0.544).

## 4.5. Test results of mediation effects

The mediation effects of social avoidance and distress (SAD) and psychological resilience (PSR) in the relationship between perceived school climate (PSC) and negative emotions (NEE) were tested using structural equation modeling (SEM). The model fit indices presented in **Table 5** indicate an excellent fit for the proposed model ($\chi^2/df$ = 2.786, CFI = 0.986, TLI = 0.982, SRMR = 0.020, RMSEA = 0.034, 90% CI = 0.028–0.041).

**Fig 3** illustrates the structural equation model, which shows the pathways from PSC to NEE through the mediating effects of SAD and PSR. All estimated paths in the model were significant at the *p* < 0.001 level, confirming the hypothesized relationships. Specifically, PSC had a direct negative effect on NEE and indirect effects mediated by SAD and PSR, with SAD and PSR also demonstrating a sequential mediating effect.

The total, direct, and indirect effects are summarized in **Table 6**. The total effect of PSC on NEE was significant (β = −0.551, *p* < 0.001), with 79.49% of this effect explained by indirect pathways. These findings confirm the central role of mediating mechanisms in the relationship between PSC and NEE.

H1: Perceived school climate is negatively associated with adolescents' negative emotions. The direct effect of PSC on NEE was significant (β = −0.113, *p* = 0.001), supporting H1. This indicates that a more positive perceived school climate directly reduces negative emotions.

H2: Social avoidance and distress mediate the relationship between perceived school climate and negative emotions. The indirect pathway PSC → SAD → NEE (β = −0.133, *p* = 0.001) was significant, confirming H2. This indirect effect accounted for 24.14% of the total effect, highlighting the critical role of SAD as a mediator.

H3: Psychological resilience mediates the relationship between perceived school climate and negative emotions. The indirect pathway PSC → PSR → NEE (β = –0.080, *p* = 0.001) was significant, supporting H3. This pathway explained 14.52% of the total effect, emphasizing the importance of PSR in mitigating negative emotions.

H4: Social avoidance and distress and psychological resilience jointly form a chain mediation effect between perceived school climate and negative emotions. The chain mediation pathway PSC → SAD → PSR → NEE (β = –0.225, *p* = 0.001) was the strongest indirect effect, accounting for 40.83% of the total effect. This finding supports H4, demonstrating that SAD and PSR work sequentially to mediate the impact of PSC on NEE.

PSC, perceived school climate. NEE, negative emotions. SAD, social avoidance and distress. PSR, psychological resilience. Boot LLCI, the lower bound of the 95% confidence interval. Boot ULCI, the upper limit of the 95% confidence interval (Percentile Bootstrap Method with Bias Correction). The Bootstrap sample size is set at 2000.

The results provide robust evidence for H1 to H4. Perceived school climate was found to directly reduce negative emotions, while its effects were largely mediated by SAD and PSR. The chain mediation mechanism further highlights the dynamic and interconnected roles of these mediators in shaping the relationship between PSC and NEE.

## 4.6 Testing for structural invariance across gender

To determine whether the structural relationships among perceived school climate (PSC), social avoidance and distress (SAD), psychological resilience (PSR), and negative emotions (NEE) differ by gender, structural invariance testing was

**Table 5. Questionnaire model fitting indicators.**

| Model fit | $\chi^2/df$ | CFI | TLI | SRMR | RMSEA (90%CI) |
|---|---|---|---|---|---|
| Model | 2.786 | 0.986 | 0.982 | 0.020 | 0.034 (0.028–0.041) |

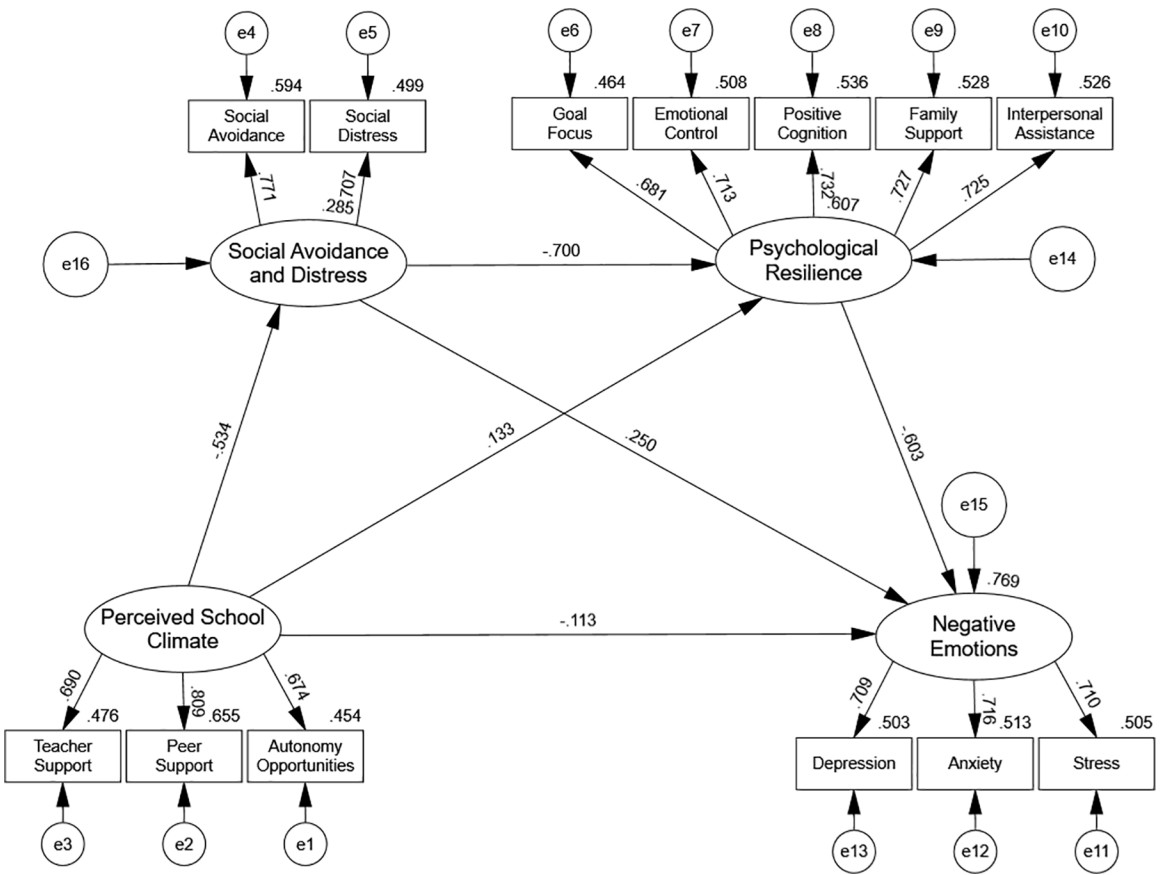

**Fig 3. Structural equation model of the relationship between perceived school climate and negative emotions, mediated by social avoidance and distress, and psychological resilience.** All paths are significant at the 0.001 level.

**Table 6. Total, direct and indirect effects in the multiple mediator model.**

| Path | Estimated effect | Boot SE | P | Boot LLCI | Boot ULCI | Ratio |
|---|---|---|---|---|---|---|
| **Direct effect** | | | | | | |
| PSC→NEE | −0.113 | 0.035 | 0.001 | −0.180 | −0.042 | 20.51% |
| **All indirect effects** | | | | | | 79.49% |
| PSC→SAD→NEE | −0.133 | 0.029 | 0.001 | −0.194 | −0.083 | 24.14% |
| PSC→PSR→NEE | −0.080 | 0.025 | 0.001 | −0.131 | −0.032 | 14.52% |
| PSC→SAD→PSR→NEE | −0.225 | 0.024 | 0.001 | −0.279 | −0.182 | 40.83% |
| **Total effect** | −0.551 | 0.027 | 0.000 | −0.603 | −0.495 | 100% |

conducted. Table 7 summarizes the fit indices for the invariance models, which include unconstrained, measurement weights, structural weights, structural covariances, and structural residuals.

The fit indices for all invariance models demonstrate excellent model fit, with $\chi^2/df < 2$, CFI > 0.980, TLI > 0.980, SRMR < 0.030, and RMSEA < 0.030. Furthermore, changes in CFI ($\Delta$CFI) and TLI ($\Delta$TLI) across successive models are all below the recommended threshold of 0.01. These results indicate that: The measurement properties are equivalent between male and female groups. The structural relationships among variables (e.g., PSC → SAD → NEE) are consistent in terms of their presence, direction, and statistical significance.

**Table 7. Testing for structural invariance across gender.**

| | χ 2/*df* | CFI | △CFI | TLI | △TLI | SRMR | RMSEA (90%CI) |
|---|---|---|---|---|---|---|---|
| Unconstrained | 1.820 | 0.987 | – | 0.983 | – | 0.024 | 0.023 (0.018–0.028) |
| Measurement weights | 1.760 | 0.987 | 0.000 | 0.984 | +0.001 | 0.025 | 0.022 (0.018–0.027) |
| Structural weights | 1.894 | 0.984 | −0.003 | 0.982 | −0.001 | 0.028 | 0.024 (0.020–0.029) |
| Structural covariances | 1.908 | 0.984 | −0.003 | 0.981 | −0.002 | 0.032 | 0.025 (0.020–0.029) |
| Structural residuals | 1.899 | 0.984 | −0.003 | 0.982 | −0.001 | 0.032 | 0.024 (0.020–0.029) |

Thus, the results confirm that the overall structure of the proposed model is invariant across genders.

## 4.7 Moderating effect of physical exercise level

To test Hypothesis 5 (H5), which posits that physical exercise level (PEL) moderates the relationship between perceived school climate (PSC) and negative emotions (NEE), a moderation analysis was conducted. The results, summarized in **Table 8**, provide clear evidence supporting H5.

The interaction term between PSC and physical exercise level (PEL) was significant (β = 0.0023, $p$ = 0.0059), indicating that PEL moderates the relationship between PSC and NEE. Specifically, the negative association between PSC and NEE becomes stronger as PEL increases. This finding suggests that higher levels of physical exercise amplify the protective effects of a positive school climate on reducing negative emotions.

The main effects of PSC and PEL were also analyzed. PSC demonstrated a significant negative association with NEE (β = −0.4272, $p$ = 0.000), indicating that a more positive perceived school climate is directly associated with fewer negative emotions. However, the main effect of PEL on NEE was non-significant (β = 0.0001, $p$ = 0.8487), suggesting that physical exercise level alone does not directly impact negative emotions but instead plays a role in moderating the effects of PSC, these findings validate H5.

The moderating effect of physical exercise level on the relationship between perceived school climate (PSC) and negative emotions (NEE) is further illustrated in **Fig 4**. The graph depicts the interaction between PSC and physical exercise level, with low, medium, and high levels of physical exercise represented as separate lines.

The results indicate that the negative association between PSC and NEE is stronger for adolescents with higher levels of physical exercise. Specifically, as perceived school climate improves, negative emotions decrease more significantly for those engaging in higher levels of physical activity, as evidenced by the steeper slope of the line representing high physical exercise level. In contrast, the line for low physical exercise level is less steep, suggesting a weaker effect of PSC on NEE for adolescents with lower physical activity.

This interaction supports the notion that physical exercise enhances the protective role of a positive school climate against negative emotions. It highlights the critical role of physical activity in amplifying the emotional benefits of a supportive school environment, particularly for adolescents at risk of negative emotional outcomes.

**Table 8. Moderation analysis of physical exercise level on the relationship between perceived school climate and negative emotions.**

| | Negative emotions | | | | |
|---|---|---|---|---|---|
| | β | SE | t | *P* | 95% CI |
| Constant | 2.5635 | 0.0133 | 192.4133 | 0.000 | (2.5373, 2.5896) |
| Perceived school climate | −0.4272 | 0.0251 | −17.0386 | 0.000 | (−0.4764, −0.3780) |
| Physical exercise level | 0.0001 | 0.0005 | 0.1908 | 0.8487 | (−0.0008, 0.0010) |
| Perceived school climate * Physical exercise level | 0.0023 | 0.0008 | 2.7577 | 0.0059 | (0.0007, 0.0040) |

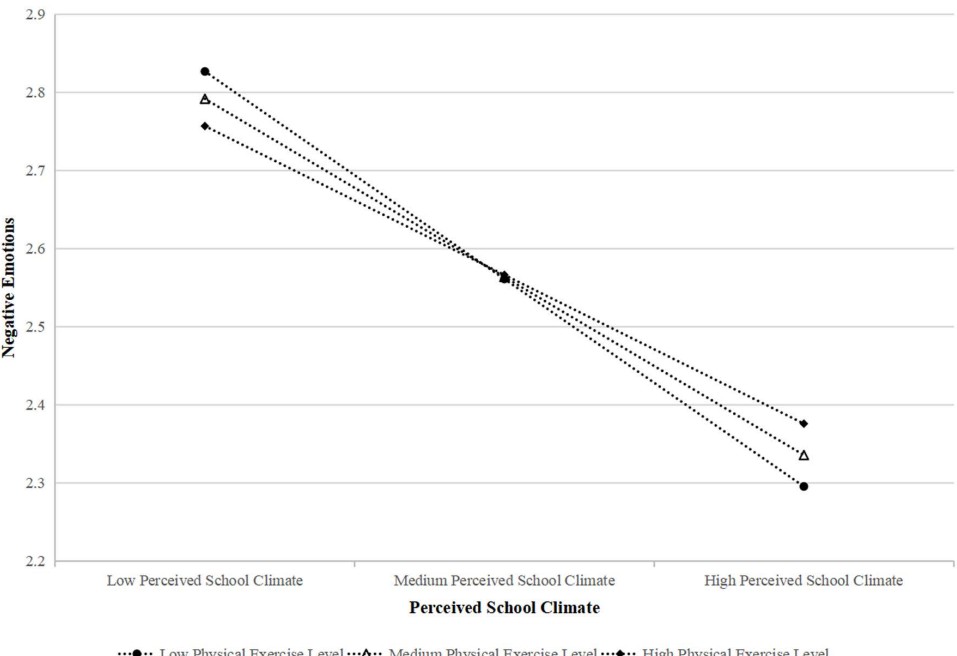

**Fig 4. Interaction effect of perceived school climate and physical exercise level on negative emotions.**

## 5. Discussion

### 5.1 Differences in key variables across demographic groups

The analysis of demographic variables revealed significant differences in PSC, SAD, PSR, NEE, and PEL across gender, grade levels, and family location. Gender differences showed that female adolescents experienced higher levels of SAD and NEE and lower levels of PSR compared to males, reflecting greater social and emotional challenges. This finding aligns with prior research highlighting gender-based disparities in emotional processing and coping mechanisms during adolescence [78,79]. Tailored interventions focusing on resilience-building and emotional regulation might particularly benefit female students, addressing their heightened vulnerability to social and emotional stressors [80].

Grade-level differences indicated a decline in PSC and PSR, coupled with an increase in NEE and SAD, as adolescents progressed to higher grades. This trend may reflect the compounding pressures of academics and social relationships in later grades. The findings underscore the need for developmental-stage-specific interventions to counteract these declines. Programs emphasizing peer support and stress management might help mitigate the rising challenges faced by older adolescents [21].

Family location emerged as another significant factor, with rural adolescents reporting higher PSC, PSR, and PEL compared to their urban counterparts. This result suggests that rural adolescents might perceive closer school-community connections and more engagement in physical activity. Urban adolescents, on the other hand, may benefit from strategies aimed at strengthening school support systems and promoting physical activity, addressing these disparities and fostering emotional well-being [81,82].

### 5.2 The direct relationship between perceived school climate and negative emotions

The results confirmed a significant negative relationship between perceived school climate (PSC) and negative emotions (NEE), supporting Hypothesis 1 (H1). Adolescents who perceive their school climate as supportive, inclusive, and

structured report lower levels of negative emotions, emphasizing the crucial role of environmental factors in shaping emotional well-being. A positive school climate, characterized by teacher and peer support and opportunities for autonomy, provides adolescents with a sense of security and belonging, which helps mitigate the psychological toll of stressors and fosters emotional stability [83,84]. The observed direct effect aligns with ecological frameworks that emphasize the role of proximal environments in adolescent development and reinforces the importance of cultivating a positive school climate as a protective factor against negative emotions [85,86].

Among the key components of a supportive school climate, teacher and peer relationships play an essential role in shaping emotional experiences. Teachers who provide warmth, encouragement, and consistent support help students feel psychologically safe and valued, which contributes to lower levels of anxiety, depression, and emotional distress. A supportive teacher-student relationship offers not only academic guidance but also emotional reassurance, helping adolescents develop adaptive coping mechanisms in response to stressors [87]. Similarly, peer support fosters social connectedness and inclusion, reducing feelings of loneliness and rejection, which are closely linked to increased emotional distress. When adolescents experience positive peer relationships, they are more likely to develop effective emotional regulation strategies, while negative peer interactions—such as exclusion, bullying, or social isolation—can exacerbate negative emotional states [88,89].

Beyond social support, perceived autonomy within the school environment contributes to emotional well-being by reinforcing a sense of agency and control [90]. When students feel that their opinions are respected and that they have opportunities to make meaningful contributions to their school life, they are more likely to experience psychological stability. Conversely, highly restrictive school environments that emphasize rigid discipline and lack of student involvement may induce feelings of helplessness and frustration, leading to increased negative emotions. While structure and guidance remain important, balancing authority with opportunities for student participation is essential in fostering a psychologically healthy school climate.

Overall, these findings underscore the significant role of the school climate in shaping adolescent emotional outcomes. A positive, inclusive, and supportive school environment functions as a protective factor that helps adolescents navigate the social and emotional challenges of this developmental stage. The alignment of these results with ecological and social support theories highlights the necessity of strengthening school climate interventions that prioritize emotional well-being alongside academic success.

### 5.3.  The mediating roles of social avoidance and distress, and psychological resilience

The mediation analyses provide strong evidence for the sequential mediating roles of social avoidance and distress (SAD) and psychological resilience (PSR) in linking perceived school climate (PSC) and negative emotional experiences (NEE), supporting Hypotheses 2 (H2), 3 (H3), and 4 (H4). Specifically, SAD was found to partially mediate the relationship between PSC and NEE, suggesting that a negative school climate fosters social withdrawal, which subsequently exacerbates emotional distress. This finding aligns with the Stress-Buffering Hypothesis, which posits that social support serves as a protective factor, while its absence intensifies maladaptive coping strategies such as avoidance and distress [60,61]. Although social withdrawal may temporarily reduce negative interactions, it limits access to emotional validation and positive peer experiences, reinforcing long-term emotional vulnerability. Consistent with prior research, chronic social avoidance has been linked to heightened risks of depression and anxiety, emphasizing its crucial role in adolescent emotional health [91,92].

Our findings further demonstrate that PSR serves as a secondary mediator, highlighting its role as an adaptive capacity shaped by prior social-emotional experiences. Adolescents in a supportive school climate are more likely to develop stronger coping skills, emotional regulation, and psychological resources, which in turn reduce negative emotions. This aligns with Developmental Cascade Theory, which emphasizes how early social experiences contribute to long-term psychological adaptation [63,64]. Importantly, while SAD significantly predicts lower resilience, the reverse pathway from PSR to

SAD is not significant, suggesting that social avoidance and distress hinder resilience development rather than resilience directly modifying social behaviors. Adolescents who frequently withdraw from social interactions due to school-related distress may miss opportunities to develop adaptive coping mechanisms, making them less equipped to manage future stressors. This finding supports our conceptualization of resilience as a mediator rather than a moderator, as it is shaped by earlier experiences rather than serving as a stable protective factor.

Prior research has often treated resilience as a moderator, suggesting that it buffers the effects of negative school experiences on emotional outcomes [93,94]. However, additional analyses testing resilience as a moderator in our model found no significant interaction effects. This suggests that resilience does not independently alter the relationship between school climate and emotional well-being but rather develops as a consequence of social experiences. These results reinforce the view that resilience is not a fixed trait but a dynamic capacity shaped by past social interactions and distress.

By demonstrating that PSC influences NEE through a sequential process involving SAD and PSR, our findings provide a more comprehensive understanding of the interplay between environmental, social, and psychological factors. The chain mediation effect suggests that the protective effects of a positive school climate on emotional well-being operate through both reduced social avoidance and enhanced resilience. This highlights the importance of considering both immediate social responses and longer-term psychological adaptation in adolescent emotional development.

Although our findings support the proposed mediation pathways, the cross-sectional nature of this study limits causal interpretations. While our theoretical framework and statistical analyses provide strong evidence for the directionality of effects, bidirectional influences may exist. Future longitudinal studies are needed to track changes in social avoidance, resilience, and emotional outcomes over time, examine whether improvements in school climate lead to corresponding shifts in social and emotional functioning, and determine whether interventions targeting social engagement enhance resilience development. Longitudinal research will be valuable in further clarifying these relationships and strengthening the evidence base for understanding how school climate influences adolescent emotional well-being.

### 5.4 The moderating role of physical exercise level

The moderating role of physical exercise in the relationship between perceived school climate (PSC) and negative emotions (NEE) provides valuable insights into the interaction between environmental and behavioral factors in adolescent emotional well-being. The results indicate that physical exercise significantly strengthens the negative association between PSC and NEE, suggesting that higher levels of physical activity enhance the protective effects of a supportive school environment on adolescents' emotional health, supporting Hypothesis 5 (H5). This finding aligns with the Stress-Buffering Hypothesis, which posits that behavioral and physiological coping mechanisms, such as exercise, can mitigate the psychological impact of external stressors [60,61].

Our findings extend previous research emphasizing the regulatory benefits of physical activity in emotional well-being. Regular physical exercise enhances mood stability, reduces stress sensitivity, and strengthens self-efficacy, which may amplify the emotional benefits of a positive school climate. In line with Developmental Cascade Theory, engaging in physical exercise reinforces adaptive psychological processes over time, thereby enhancing adolescents' ability to cope with environmental stressors [63,64]. In this study, adolescents engaging in higher levels of physical exercise experienced a greater reduction in negative emotions when they perceived a more positive school climate, as evidenced by the steeper slope of the interaction effect for high exercise levels.

Interestingly, physical exercise alone did not exert a direct main effect on NEE, suggesting that its protective role depends on environmental context rather than functioning as an independent determinant of emotional health. This indicates that while exercise may enhance emotional resilience, its benefits are more pronounced when combined with a positive school climate. For adolescents with low levels of physical activity, the advantages of a favorable school climate were less pronounced, emphasizing that behavioral and environmental factors work synergistically rather than independently.

These findings highlight the importance of integrating behavioral interventions with school-based strategies to promote adolescent well-being. Schools and policymakers should prioritize holistic approaches that encourage physical activity alongside efforts to enhance school climate, as such combined strategies may offer a more comprehensive framework for mitigating emotional challenges among adolescents, particularly for those at heightened risk of negative emotional outcomes.

## 6. Implications for educational management and policy

### 6.1. Fostering a positive school climate

The direct association between perceived school climate (PSC) and negative emotions (NEE) highlights the importance of creating supportive and inclusive school environments. Policymakers should prioritize investments in teacher training programs that emphasize the role of emotional support and mentorship. Peer-support initiatives, such as peer mentoring or group-based activities, can further foster a sense of belonging and reduce social isolation among students. Additionally, schools should provide opportunities for student autonomy, encouraging active participation in decision-making processes, which has been shown to enhance emotional well-being.

### 6.2. Promoting social connectedness and reducing avoidance

Given the mediating role of social avoidance and distress (SAD), school interventions should focus on promoting positive peer interactions and reducing social withdrawal. Anti-bullying campaigns, conflict-resolution programs, and inclusive extracurricular activities can help reduce SAD by cultivating a safe and welcoming school environment. Teachers and counselors can also be trained to identify early signs of social avoidance and implement targeted interventions. Furthermore, fostering a school culture that values diversity and inclusiveness can help mitigate the social challenges that contribute to emotional distress.

### 6.3. Building psychological resilience

Psychological resilience (PSR) emerged as a critical factor in reducing NEE, emphasizing the need for resilience-building programs within schools. Policymakers should encourage the integration of resilience training into the curriculum, focusing on skills such as emotional regulation, problem-solving, and coping strategies. Mindfulness and stress-management workshops can be incorporated into regular school schedules to provide students with tools to navigate challenges effectively. Family involvement in resilience-building programs can further reinforce these skills in home environments, creating a supportive network for adolescents.

### 6.4. Encouraging physical activity

The moderating effect of physical exercise level (PEL) underscores its potential to enhance the emotional benefits of a positive school climate. Schools should provide diverse opportunities for physical activity, including structured sports programs, recreational activities, and active breaks during the school day. Policymakers can also incentivize schools to improve access to physical activity facilities, particularly in underserved urban areas where participation levels may be lower. Additionally, health education programs should emphasize the psychological benefits of regular exercise, encouraging students to adopt physically active lifestyles as a means of emotional regulation.

### 6.5. Addressing demographic disparities

The observed differences in key variables across gender, grade levels, and family locations suggest that tailored interventions are essential to address the unique needs of different demographic groups. Gender-specific programs can be implemented to address the heightened vulnerability of female adolescents to SAD and NEE, while resilience-building efforts may be more effective for females. For older adolescents, schools should focus on mitigating the academic and

social pressures that contribute to declines in PSC and PSR. Urban schools, in particular, may need targeted support to enhance school-community connections and provide accessible physical activity options.

## 6.6. Evidence-based decision-making

Finally, the findings of this study underscore the importance of evidence-based decision-making in educational management. Policymakers and school administrators should rely on data-driven approaches to assess the effectiveness of interventions and make iterative improvements. Longitudinal monitoring of school climate, student well-being, and participation in physical activity can provide valuable insights into the impact of implemented programs. Additionally, partnerships with researchers and mental health professionals can facilitate the development and evaluation of innovative interventions.

## 7. Limitations and future directions

### 7.1. Limitations

**7.1.1. Cross-sectional design.** This study employed a cross-sectional design, which restricts our ability to establish causal relationships among variables. While our findings align with theoretical frameworks and previous research, the directionality of effects remains uncertain. For instance, while we hypothesize that perceived school climate (PSC) influences negative emotional experiences (NEE) through social avoidance and distress (SAD) and psychological resilience (PSR), it is also possible that pre-existing emotional distress influences students' perceptions of their school climate and social interactions.

To address this limitation, longitudinal studies are needed to examine temporal changes in these variables and confirm the hypothesized pathways. Additionally, experimental interventions that improve school climate and track subsequent emotional and behavioral outcomes could help establish stronger causal inferences regarding the protective role of PSC and PSR in adolescent mental health.

**7.1.2. Self-reported data.** This study relied solely on self-reported data, which may introduce social desirability bias, recall inaccuracies, and common method bias. Adolescents may underreport negative emotions due to social expectations or personal denial, while overestimating resilience based on self-perception biases. Moreover, subjective reports on social avoidance and distress (SAD) may not fully capture behavioral manifestations, limiting the objectivity of our findings.

To mitigate self-report biases, we implemented several control measures, including ensuring participant anonymity, using validated psychological scales, and emphasizing response honesty during the survey process (see Section 3.1.3). However, future research should incorporate multi-informant data sources, such as teacher reports, peer assessments, or observational methods, to provide a more comprehensive and objective evaluation of adolescents' emotional and behavioral experiences. Additionally, biometric or behavioral tracking tools (e.g., wearable activity monitors for physical exercise) could enhance measurement reliability.

**7.1.3. Cultural and contextual specificity.** The study was conducted within a specific cultural and educational context, which may limit its generalizability to other settings. For example, differences in cultural values, school policies, and societal norms could influence the relationships between PSC, SAD, PSR, and NEE. Future studies should examine these variables in diverse cultural and educational contexts to explore potential cross-cultural variations.

**7.1.4. Limited scope of physical exercise analysis.** Although PEL was identified as a moderator, this study did not examine the specific characteristics of physical activity, such as intensity, frequency, or type. Different forms of exercise (e.g., team sports vs. individual activities) may have varying effects on emotional outcomes, and future research could delve into these distinctions to offer more nuanced insights.

**7.1.5. Exclusion of additional influential factors.** The study focused on PSC, SAD, PSR, and PEL as key variables but did not account for other potentially relevant factors, such as socioeconomic status, parental support, or digital media use, which may also influence adolescent emotional outcomes. Including these variables in future studies could provide a more comprehensive understanding of the multifaceted influences on adolescent well-being.

### 7.2 Future directions

**7.2.1. Longitudinal and experimental research.** Future studies should adopt longitudinal designs to track changes in PSC, SAD, PSR, and NEE over time. Experimental interventions, such as implementing school climate improvement programs or physical activity initiatives, could provide causal evidence and evaluate the effectiveness of targeted strategies.

**7.2.2. Multimodal assessments.** To enhance the validity of findings, future research should incorporate multimodal assessments, such as physiological measures (e.g., stress biomarkers), behavioral observations, and qualitative interviews. These approaches can complement self-reported data and provide a more holistic view of adolescent emotional health.

**7.2.3. Cross-cultural comparisons.** Conducting comparative studies across different cultural and educational contexts would help identify universal versus culture-specific pathways linking PSC and NEE. Such research could guide the adaptation of interventions to fit the unique needs of diverse populations.

**7.2.4. Detailed analysis of physical activity.** Future research should investigate the differential effects of various forms, intensities, and durations of physical activity on emotional outcomes. Exploring the mechanisms through which exercise moderates the PSC-NEE relationship, such as neurochemical changes or social bonding opportunities, could deepen our understanding of its protective role.

**7.2.5. Integration of additional factors.** Expanding the theoretical framework to include variables such as family dynamics, peer influence, and digital media use could enrich our understanding of adolescent well-being. These factors may interact with PSC, SAD, PSR, and PEL, creating complex networks of influence that warrant further exploration.

**7.2.6. Gender-specific interventions.** Given the observed gender differences in key variables, future research should explore gender-specific mechanisms in greater depth. Interventions tailored to the unique needs of male and female adolescents could enhance their effectiveness and ensure equitable support.

## 8. Conclusion

This study highlights the significant demographic disparities in key variables, revealing gender, grade-level, and family location differences in perceived school climate, social avoidance and distress, psychological resilience, and negative emotions. These findings emphasize the importance of tailoring interventions to address the unique needs of diverse adolescent populations. Additionally, the research confirms the critical role of perceived school climate in reducing negative emotions both directly and indirectly through social avoidance, distress, and psychological resilience. The moderating role of physical exercise further underscores its value in enhancing the protective effects of a supportive school environment. Together, these findings provide a comprehensive framework to inform theory and guide practical interventions aimed at promoting emotional well-being and resilience among adolescents in varied educational and social contexts.

## Supporting information

**S1 File. Empirical data.**
(XLSX)

**S2 File. Statistical analysis results.**
(ZIP)

## Acknowledgments

The authors would like to extend their heartfelt thanks to Chengdu Tianfu No. 7 High School for their invaluable support during this research. We also express our sincere gratitude to all the adolescents who participated in the survey. Special thanks are extended to our colleagues at the School of Foreign Languages, Chengdu Sport University, for their insightful suggestions on improving the language and expression of this study.

## Author contributions

**Conceptualization:** Weisong Chen.

**Data curation:** Zhen Huang.

**Funding acquisition:** Weisong Chen.

**Investigation:** Weisong Chen, Bo Peng.

**Methodology:** Zhen Huang.

**Project administration:** Bo Peng.

**Supervision:** Haibo Yan.

**Validation:** Haibo Yan.

**Visualization:** Zhen Huang.

**Writing – original draft:** Weisong Chen.

**Writing – review & editing:** Weisong Chen, Zhen Huang, Bo Peng, Haibo Yan.

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
