## [Decision Letter · Decision Letter 0]

14 Jan 2025

PONE-D-24-57337Exploring the Mechanisms Linking Perceived School Climate to Negative Emotions in Adolescents: The Mediating Roles of Social Avoidance and Distress, and Psychological Resilience, with Physical Exercise Level as a ModeratorPLOS ONE

Dear Dr. Yan,

Thank you for submitting your manuscript to PLOS ONE. After careful consideration, we feel that it has merit but does not fully meet PLOS ONE’s publication criteria as it currently stands. Therefore, we invite you to submit a revised version of the manuscript that addresses the points raised during the review process.

We look forward to receiving your revised manuscript.

Kind regards,

Henri Tilga, PhD

Academic Editor

PLOS ONE

**Journal Requirements:**

2. In the ethics statement in the Methods, you have specified that verbal consent was obtained. Please provide additional details regarding how this consent was documented and witnessed, and state whether this was approved by the IRB.

This study was supported  by the Major Project of the Sichuan Key Research Base of Philosophy and Social Sciences (Grant No. SC21EZD006), titled “Research on the Collaborative Development of School Physical Education in the Chengdu-Chongqing Economic Circle from the Perspective of Integration of Sports and Education.” The funders had no role in the study design, data collection and analysis, decision to publish, or preparation of the manuscript.

Reviewers' comments:

Reviewer's Responses to Questions

**Comments to the Author**

1. Is the manuscript technically sound, and do the data support the conclusions?

Reviewer #1: Yes

2. Has the statistical analysis been performed appropriately and rigorously? 

Reviewer #1: Yes

3. Have the authors made all data underlying the findings in their manuscript fully available?

Reviewer #1: Yes

4. Is the manuscript presented in an intelligible fashion and written in standard English?

Reviewer #1: Yes

5. Review Comments to the Author

**Reviewer #1: ** This study is cross-sectional nature and all data were collected from the sole self-reports. This would cause some problems when explaining the direction of the chain mediating effects. There are many references I cannot find and I suppose those were published in Chinese journals and thus not able to judge the quality. However, there were measurements for some key variables. I am also wondering whether resilience could be changed via social avoidance and social distress. From my experience, there are many previous studies with resilience as moderator. Robert Coplan also found that sport exercise appeared to be a mediator between shyness and adjustment outcomes. Why the authors chose physical exercise as moderator? Therefore, the conceptual model needs to be reconsidered and a stronger rationale would be more convincing.

6. PLOS authors have the option to publish the peer review history of their article (what does this mean? ). If published, this will include your full peer review and any attached files.

**Do you want your identity to be public for this peer review?** For information about this choice, including consent withdrawal, please see our Privacy Policy .

Reviewer #1: No

---

## [Author Response · Author response to Decision Letter 1]

6 Mar 2025

Response to Reviewer 1

Comment 1:

This study is cross-sectional nature and all data were collected from the sole self-reports. This would cause some problems when explaining the direction of the chain mediating effects.

Response:

We acknowledge that our cross-sectional design and sole reliance on self-reported data limit our ability to definitively establish causal direction in the proposed chain mediation model. In response to this concern, we have made the following revisions:

(1) Addition to Section 3.1.3 (Data Collection Methods – Control Measures to Reduce Bias): We have added a new subsection in Section 3.1.3 to describe in detail the measures implemented to minimize biases associated with self-reports. The revised text includes:

① Anonymity Assurance: Participants were explicitly informed that their responses would remain anonymous and confidential, reducing social desirability pressures.

② Validated Psychological Scales: All measures used were selected based on previous validations in adolescent populations, ensuring high reliability and construct validity.

③ Balanced Question Framing: Items were carefully worded to be neutral and non-leading.

④ Standardized Instructions: Both verbal and written instructions emphasized the importance of honest and accurate responses.

⑤ Response Consistency Checks: Post-survey screening was conducted to detect inconsistent or extreme responses.

⑥ Missing Data Handling: All samples with missing data were completely removed from the dataset to maintain data integrity.

(2) Enhancement of Section 2.5 (Theoretical Foundations): To bolster our conceptual model and address concerns about directional causality, we have substantially revised Section 2.5. In this section, we integrate two theoretical frameworks—the Stress-Buffering Hypothesis and Developmental Cascade Theory—to explain the proposed sequence of mediators. Specifically:

① The Stress-Buffering Hypothesis (Cohen & Wills, 1985) supports our contention that a negative school climate triggers immediate social avoidance and distress (SAD) as a maladaptive coping response.

② The Developmental Cascade Theory (Masten & Cicchetti, 2010) provides the rationale for the subsequent development (or inhibition) of psychological resilience (PSR) as a consequence of prolonged social distress.

③ We have also clarified the rationale for including physical exercise (PEL) as a moderator, emphasizing its role as a modifiable behavioral factor that can either amplify the benefits of a positive school climate or buffer against the adverse effects of a negative one.

(3) Revisions to Section 7.1 (Limitations): In Section 7.1, we have explicitly acknowledged two key limitations:

① Cross-Sectional Design: We now state that the cross-sectional nature restricts causal inference. We explain that, although our findings are consistent with theoretical expectations, longitudinal or experimental research is needed to confirm the temporal ordering of effects.

② Self-Reported Data: We note that relying solely on self-reports may introduce biases (e.g., social desirability, recall inaccuracies) and that future research should incorporate multi-informant sources and objective measures (such as wearable activity monitors) to enhance validity.

We believe these revisions address the reviewer’s concerns by providing a more transparent discussion of methodological limitations and by offering a stronger theoretical foundation to support our hypothesized model. We appreciate the reviewer’s suggestions, which have significantly improved the clarity and rigor of our study.

Comment 2:

There are many references I cannot find and I suppose those were published in Chinese journals and thus not able to judge the quality.

Response:

Thank you for raising this concern. In our revised manuscript, we have made every effort to ensure that references are accessible. Unless unavoidable, we have replaced references from Chinese journals with corresponding literature from international journals to facilitate ease of access and evaluation. We appreciate your feedback, which has helped us enhance the transparency and quality of our reference list.

Comment 3:

However, there were measurements for some key variables. I am also wondering whether resilience could be changed via social avoidance and social distress. From my experience, there are many previous studies with resilience as moderator. 

Response:

We sincerely appreciate the reviewer’s insightful comment regarding the role of psychological resilience. Your observation that many previous studies have examined resilience as a moderator is valuable, and it prompted us to carefully reconsider our model from both theoretical and empirical perspectives. In response, we conducted additional analyses and refined our discussion to further strengthen our justification for conceptualizing resilience as a mediator in our study.

(1) Theoretical Justification for Resilience as a Mediator

The decision to position psychological resilience (PSR) as a mediator is rooted in two well-established theoretical frameworks:

① The Stress-Buffering Hypothesis (Cohen & Wills, 1985) suggests that adverse environmental conditions—such as a negative perceived school climate (PSC)—immediately trigger stress responses, often manifesting as social avoidance and distress (SAD). Adolescents experiencing a lack of teacher support, peer exclusion, or limited autonomy tend to withdraw socially, which increases emotional distress. These immediate social-emotional responses are critical precursors to later psychological adaptations, making SAD an essential early-stage factor in the mediation process.

② The Developmental Cascade Theory (Masten & Cicchetti, 2010) explains how early social experiences influence the gradual development of adaptive capacities, including resilience. Adolescents who experience persistent social distress and avoidance have fewer opportunities to develop effective coping mechanisms, resulting in weakened resilience over time. This perspective suggests that resilience is not an inherent buffer that preemptively protects against negative experiences but rather a dynamic capacity shaped by prior social interactions.

From this theoretical standpoint, positioning resilience as a moderator (rather than a mediator) would imply that resilience exists independently of prior social experiences and immediately mitigates the impact of PSC on negative emotions (NEE). However, based on the developmental nature of resilience, we propose that it is more accurately conceptualized as an outcome of prior social experiences, which then influences emotional well-being.

(2) Empirical Support from Our Findings

In response to the reviewer’s suggestion, we tested an alternative model in which resilience served as a moderator between perceived school climate and negative emotions. However, the interaction effects were not statistically significant, suggesting that resilience does not function as a direct buffer in this relationship.

Additionally, while the pathway from social avoidance and distress to psychological resilience was statistically significant, the reverse pathway—from resilience to social avoidance and distress—was not significant. This further supports our hypothesis that resilience does not directly shape social behaviors but is instead shaped by them. In other words, adolescents’ resilience levels are influenced by their earlier social-emotional experiences, rather than resilience itself influencing their likelihood of engaging in social avoidance and distress.

(3) Enhancements to the Discussion Section

In light of these findings, we have substantially revised our discussion section to offer a more in-depth analysis of the mechanisms underlying the role of resilience. Specifically:

① We now explicitly acknowledge that prior research has treated resilience as a moderator, and we discuss the conditions under which it may serve different roles in different contexts.

② We provide a broader theoretical justification for why resilience is best conceptualized as a mediator in our specific model.

③ We discuss the implications of our empirical findings, emphasizing how resilience develops in response to prior social avoidance and distress, rather than independently moderating the impact of school climate on emotional outcomes.

④ We propose that longitudinal research would be valuable in further clarifying these relationships over time.

We sincerely appreciate the reviewer’s insightful feedback, which has helped us to critically evaluate and refine our conceptual framework. This process has strengthened our theoretical justification, enhanced the clarity of our discussion, and ultimately improved the overall rigor of our manuscript. We hope that these revisions adequately address your concerns, and we welcome any further suggestions that may help refine our work.

Thank you again for your thoughtful input.

Comment 4:

Robert Coplan also found that sport exercise appeared to be a mediator between shyness and adjustment outcomes. Why the authors chose physical exercise as moderator? Therefore, the conceptual model needs to be reconsidered and a stronger rationale would be more convincing.

Response:

Thank you for your insightful comment regarding the conceptual role of physical exercise in our model. We acknowledge that prior research, including Robert Coplan’s work, has examined physical exercise as a mediator, particularly in its role linking shyness to adjustment outcomes. These studies suggest that engaging in physical activity may serve as a behavioral mechanism through which individuals regulate emotions and facilitate social adaptation. However, in our study, we conceptualized physical exercise as a moderator rather than a mediator, based on both theoretical reasoning and empirical considerations, and we have further refined our discussion in the literature review and discussion sections to clarify this rationale.

Our conceptualization is primarily guided by the Stress-Buffering Hypothesis, which posits that certain behaviors or external factors can reduce the psychological impact of stressors. In our model, physical exercise does not function as a pathway through which school climate influences emotional outcomes, but rather as a protective factor that alters the strength of this relationship. Adolescents who engage in higher levels of physical activity may experience enhanced physiological stress resilience, improved emotion regulation, and greater self-efficacy, thereby buffering the negative effects of an unfavorable school climate. Conversely, for those with lower levels of physical activity, the detrimental effects of a negative school climate on emotional well-being may be more pronounced, as they lack this behavioral mechanism to counteract stress.

Additionally, Developmental Cascade Theory supports the notion that the protective effects of physical exercise do not necessarily emerge as a consequence of perceived school climate, but rather function as an independent behavioral resource that interacts with environmental influences over time. Adolescents with high levels of physical exercise are likely to have developed better emotion regulation skills and stress-coping mechanisms, allowing them to resist or attenuate the impact of a negative school climate. This process is not necessarily sequential as a mediation model would imply, but rather contingent on an individual’s engagement in physical activity at a given time, making it more appropriate to conceptualize exercise as a moderator rather than a mediator.

If physical exercise were positioned as a mediator, it would imply that perceived school climate directly influences adolescents’ engagement in physical activity, which in turn affects their emotional well-being. However, prior research suggests that participation in physical exercise is shaped by multiple factors beyond school climate alone, including individual motivation, parental encouragement, socioeconomic status, and access to exercise facilities. While school climate may contribute to adolescents' attitudes toward physical activity, it is unlikely to be the primary determinant of exercise behavior, making a mediating role less theoretically appropriate in this context.

To ensure greater clarity, we have expanded and refined the literature review and discussion sections to further support our conceptual framework. Specifically, we have:

(1) Strengthened our theoretical justification by elaborating on how physical exercise fits within the Stress-Buffering Hypothesis and Developmental Cascade Theory.

(2) Addressed alternative conceptualizations of physical exercise in prior research, explicitly acknowledging why it has sometimes been treated as a mediator while explaining why a moderating role is more appropriate in our study.

(3) Expanded our discussion section to provide a deeper analysis of how physical exercise interacts with school climate to influence adolescent emotional well-being, reinforcing the importance of considering behavioral regulation as a moderating mechanism.

We appreciate this valuable suggestion, as it has allowed us to further refine and justify our theoretical framework. We hope our revisions adequately address this concern, and we welcome any further feedback.

---

## [Decision Letter · Decision Letter 1]

11 Mar 2025

Exploring the mechanisms linking perceived school climate to negative emotions in adolescents: the mediating roles of social avoidance and distress, and psychological resilience, with physical exercise level as a moderator

PONE-D-24-57337R1

Dear Dr. Yan,

We’re pleased to inform you that your manuscript has been judged scientifically suitable for publication and will be formally accepted for publication once it meets all outstanding technical requirements.

Kind regards,

Henri Tilga, PhD

Academic Editor

PLOS ONE

Additional Editor Comments (optional):

Reviewers' comments:

Reviewer's Responses to Questions

**Comments to the Author**

1. If the authors have adequately addressed your comments raised in a previous round of review and you feel that this manuscript is now acceptable for publication, you may indicate that here to bypass the “Comments to the Author” section, enter your conflict of interest statement in the “Confidential to Editor” section, and submit your "Accept" recommendation.

Reviewer #1: All comments have been addressed

2. Is the manuscript technically sound, and do the data support the conclusions?

Reviewer #1: Yes

3. Has the statistical analysis been performed appropriately and rigorously? 

Reviewer #1: Yes

4. Have the authors made all data underlying the findings in their manuscript fully available?

Reviewer #1: Yes

5. Is the manuscript presented in an intelligible fashion and written in standard English?

Reviewer #1: Yes

6. Review Comments to the Author

Reviewer #1: (No Response)

7. PLOS authors have the option to publish the peer review history of their article (what does this mean? ). If published, this will include your full peer review and any attached files.

**Do you want your identity to be public for this peer review?** For information about this choice, including consent withdrawal, please see our Privacy Policy .

Reviewer #1: No

---

## [Editor Report · Acceptance letter]

PONE-D-24-57337R1

PLOS ONE

Dear Dr. Yan,

I'm pleased to inform you that your manuscript has been deemed suitable for publication in PLOS ONE. Congratulations! Your manuscript is now being handed over to our production team.

Kind regards,

on behalf of

Dr. Henri Tilga

Academic Editor

PLOS ONE